# Effects of Aquatic Exercises for Patients with Osteoarthritis: Systematic Review with Meta-Analysis

**DOI:** 10.3390/healthcare10030560

**Published:** 2022-03-16

**Authors:** Ji-Ah Song, Jae Woo Oh

**Affiliations:** Department of Nursing, Konyang University, Daejeon 35365, Korea; jasong@konyang.ac.kr

**Keywords:** osteoarthritis, hydrotherapy, exercise therapy, systematic review

## Abstract

This systematic review examined papers published in Korean, English and newer publications that were not included in previous studies to assess the effect size of aquatic exercise-based interventions on pain, quality of life and joint dysfunction among patients with osteoarthritis. Six national and international databases were used to review literature (published up to 7 March 2019) on randomized controlled trials of aquatic exercise-based interventions in patients with osteoarthritis. For the 20 studies included, a meta-analysis showed that aquatic exercise produces 0.61-point reduction (*n* = 756; mean difference (MD) = −0.61; 95% confidence interval (CI): −0.90–−0.32) in pain compared with a control group, and aquatic exercise was effective in reducing pain (*n* = 315; MD = −0.28; 95% CI: −0.50–−0.05) compared with a land-based exercise group. Another meta-analysis showed that aquatic exercise produces 0.77-point improvement in quality of life (*n* = 279; MD = −0.77; 95% CI: −1.38–−0.15) compared with a control group. Finally, a meta-analysis showed that aquatic exercise produces 0.34-point reduction in joint dysfunction (*n* = 279; MD = −0.77; 95% CI: −1.38–−0.15) compared with a control group. For patients with osteoarthritis, aquatic-exercise-based interventions are effective for reducing pain and joint dysfunction and improving quality of life.

## 1. Introduction

Osteoarthritis is an aging-related chronic degenerative disease that begins at the age of approximately 50–60 years [1] and commonly affects the knee and hip joints [2]. The main symptom of osteoarthritis is pain; however, joint stiffness, instability and weakness are also observed; these symptoms result in functional limitations in daily life, difficulty performing activities, lower quality of life and systemic inflammation [1,3,4,5].

The Korea Centers for Disease Control and Prevention describe osteoarthritis as a representative musculoskeletal disease caused by aging and has reported that its prevalence among the entire Korean population is 12.5%, and 5.1% and 18.9% among men and women aged over 50 years, respectively [2]. In particular, osteoarthritis is observed in three out of 10 women over the age of 70 and most commonly presents in the knee area (36.1%) [2].

Treatment for osteoarthritis includes drug therapy, surgical therapy and exercise [3,6]. In addition, massage, aromatherapy, hot therapy and cold therapy are administered to relieve symptoms [6]. Among these treatments, exercise is particularly highly recommended because it is easy to perform, inexpensive and has a low risk of side effects [3]. The most commonly recommended types of exercise for patients with osteoarthritis include muscle-strengthening movements, aerobics and exercises performed in water or on the floor that have a low impact on joints [7].

Strengthening the muscles around the joints is an important element of the treatment for osteoarthritis, as it helps to improve cartilage quality, nerve activation and coordination between muscles [8]. Moreover, stronger muscles help the joints to absorb the body weight while walking [8]; more specifically, strong muscles facilitate the absorption and distribution of weight at the hip and knee joints, thereby increasing stability and improving function and mobility [9,10].

Aquatic exercises are particularly beneficial for overweight and older patients with osteoarthritis. The waves and buoyancy of water support the weight of the body, reducing the impacts on joints and the intensity of perceived pain [11]. Furthermore, the temperature and water pressure of warm water relax muscles, relieve stress, reduce muscle stiffness, and facilitate movement [11,12,13,14]. Aquatic exercises are also effective for increasing muscle strength [15].

Many studies have reported the effects of aquatic exercise for patients with musculoskeletal problems, finding these exercises to be beneficial for pain, function, and quality of life [11,13,16]. A systematic review of the literature on the effects of aquatic exercise for patients with osteoarthritis has also been performed; however, the review only examined the effects on muscle strength, and the effect size was not considered [17]. Therefore, the present study aimed to systematically review papers published in Korean and recently published papers that have not been included in previous studies to assess the effect, as well as the effect size, of aquatic exercise on pain, joint function, and quality of life among patients with osteoarthritis. Based on the findings, this study suggests an evidence-based intervention plan for nursing practice and makes recommendations for future research regarding aquatic exercise interventions for patients with osteoarthritis.

## 2. Methods

### 2.1. Study Design

The material and methods are based on the PRISMA (Preferred Reporting Items for Systematic Reviews and Meta-Analyses) guidelines. This study comprises a systematic review of randomized controlled trials (RCTs), non-randomized clinical studies, non-comparative studies and case reports were excluded. The researchers submitted the research proposal to the institutional review board of the Konyang University based on the regulations on research ethics and collected data after obtaining review exemption approval (KYU-2019-241-01).

### 2.2. Eligibility Criteria

All study characteristics used to decide whether a study was eligible for inclusion in the systemic literature review was selected based on the PICO (Patients, Intervention, Comparator, Outcome) framework [18].

#### 2.2.1. Patients

We sought studies that featured patients who were diagnosed with osteoarthritis and who were capable of communicating.

#### 2.2.2. Intervention

Studies that administered aquatic exercise-based interventions were selected. Aquatic exercise was defined as exercises performed in water that was between waist and chest height. The exercise types were cardiovascular exercises, such as walking, running, and other movements and weight training exercises, such as stretching, dumbbell use and leg lifts [19].

#### 2.2.3. Comparator

To include comparative interventions, we sought studies that featured a control group that did not receive the experimental treatment, and/or a group that performed land-based exercises.

#### 2.2.4. Outcome

The outcome focused on in this review was the effects of aquatic exercise on patients with osteoarthritis of the lower extremities. Pain was selected as the primary outcome. Pain was measured on various scales (Visual Analog Scale; VAS, Western Ontario and McMaster Universities OA Index; WOMAC, Knee injury and Osteoarthritis Outcome Score; KOOS, etc.), and in a study using two or more pain scales, WOMAC and KOOS results were used for meta-analysis. Joint function, which is influenced by the degree of stiffness and movement of one’s joints, and quality of life, which reflects an individual’s general well-being in daily life, were analyzed as secondary outcomes.

#### 2.2.5. Inclusion and Exclusion Criteria

The inclusion criteria for articles to be systemically analyzed were as follows: (1) concerning aquatic exercise among patients with osteoarthritis, (2) featuring an RCT design, (3) published in English or Korean and (4) published in a journal.

The exclusion criteria were as follows: (1) a hydrotherapy-focused study, such as balneotherapy using mineral water, spa therapy (focusing on water temperature and flow rate), or Kneipp hydrotherapy (focusing on temperature and water pressure); (2) an animal-focused or pre-clinical study; (3) a non-comparative study; (4) an unpublished thesis; (5) a non-experimental study, such as observational research or a review; and (6) a study in which the intervention outcomes concerned cost.

### 2.3. Information Sources, Search Strategy and Selection Process

#### 2.3.1. Information Sources

For the data search, all searchable items published before 7 May 2019 were examined. International databases used for the search were PubMed, CINAHL and the Cochrane Central Register of Controlled Trials (CENTRAL). Korean databases, including the Research Information Service System, the Korean Studies Information Service System and DBpia were also searched for published journal articles. To increase the sensitivity of the literature search, gray journals (i.e., journals that are not part of traditional commercial or academic publishing and distribution channels) were searched manually using Google Scholar, and no limits were placed on the intervention outcomes. MeSH terms and text words were included among the search terms using “AND/OR” and truncation. First, intervention methods featuring the terms “‘water’ [Mesh]”, “aqua*”, “spa” and “‘exercise therapy’ [MeSH]” were searched. For participants, studies featuring the terms “‘arthritis [MeSH]” were searched. Ultimately, the search strategy was “((‘water’ [MeSH] OR ‘aqua*’ OR ‘spa’) AND ‘exercise therapy’ [MeSH] AND ‘arthritis’ [MeSH]).”

#### 2.3.2. Search Strategy and Selection Process

The studies returned through the search were filtered based on the search strategy and the inclusion and exclusion criteria. A PRISMA flow chart was created to describe the literature-selection process in detail [16]. A total of 445 articles were returned through the search. Of these, 131 articles were duplicates and were excluded. Thus, the remaining 314 articles were reviewed by two researchers, focusing on the titles and abstracts, to determine their agreement with the search strategy and selection criteria. Through this process, 205 studies were omitted, as they were not related to the search strategy, were not published in Korean or English, and/or had a study design that did not meet the selection criteria. The original texts of the remaining 109 studies were reviewed by applying the same criteria and process as that used for the titles and abstracts. Therefore, 20 articles, comprising the remaining 17 articles and three articles found through manual searches, were qualitatively analyzed. Quantitative meta-analysis was performed on 12 of these studies; in these 12 studies, the outcome variables included pain, quality of life and/or joint function in patients with osteoarthritis, and the results were presented in the form of means and standard deviations (Figure 1).

### 2.4. Quality Assessment of Articles

In this study, a critical review of the literature was conducted using the Cochrane’s Risk of Bias (RoB) tool; a quality assessment tool for RCTs [18]. The tool assesses the quality of RCTs by focusing on seven items: random sequence generation, allocation concealment, blinding of participants and personnel, blinding of outcome assessment, incomplete outcome data, selective reporting and other bias. For each item, the degree of risk is evaluated as low, uncertain or high. For the present research, quality evaluation of the selected articles was conducted independently by two researchers. If there were any inconsistencies between the researchers’ determinations, a third researcher’s opinion was obtained, and discussions were held until a consensus was reached.

### 2.5. Data Analysis

For the selected studies, systematic confirmation, synthesis, statistical merging, and reporting of the outcomes were analyzed based on the Cochrane guidelines [18], and the effect size was assessed through meta-analysis.

#### 2.5.1. Data Extraction

After analyzing the characteristics of the 20 included papers, the articles were coded and organized (Table 1).

#### 2.5.2. Selection of the Analysis Model

In this study, as a result of the data extraction methods applied, we felt that it was likely that there was heterogeneity between the studies regarding the methods, times and durations of the interventions. Thus, the data were analyzed using a random effects model [18,20].

#### 2.5.3. Effect Size Calculation

In this study, effect size was analyzed using means and standard deviations, since the target outcome variables were continuous variables. When, across two or more studies, the same measurement tool was used to measure the same intervention outcomes, the mean difference (MD) was calculated by using the final mean values provided for each intervention and control group. When various measurement tools were used across studies, the standardized MD was calculated. The effect of each outcome variable and the 95% confidence interval (CI) were analyzed using the inverse variance method.

#### 2.5.4. Heterogeneity Test

Heterogeneity refers to the differences between the individual studies included in the meta-analysis. In this study, the heterogeneity between studies was evaluated using Higgins I^2^-statistic. When the I^2^ value is 25%, 50% or above 75%, the heterogeneity is considered low, moderate or high, respectively [18,20].

## 3. Results

### 3.1. General Characteristics of the Systemically Reviewed Articles

The general characteristics of the 20 studies [14,21,22,23,24,25,26,27,28,29,30,31,32,33,34,35,36,37,38,39] included in the systematic literature review are as follows (Table 1). All patients included in the studies were aged 48 years or older. Two studies focused on patients with hip osteoarthritis (10%), eight studies featured patients with knee osteoarthritis (40%), six studies featured patients with hip and knee osteoarthritis (30%) and four studies (20%) did not limit their patient sample to any particular type of osteoarthritis.

The aquatic exercise programs were administered by trained experts in pools in which the water was between waist and chest height and was 28–34 °C. The most common durations of the intervention sessions were 50 and 60 min, respectively; these durations were used by seven studies each. The number of sessions per week varied from two to five. The most common duration of the intervention program was 12 weeks (six studies), and the range across the studies analyzed was 4–24 weeks.

All 20 studies included in the analysis measured joint function to assess the effect of aquatic exercise. Thirteen studies (65%) measured pain using a visual analog scale and a self-report questionnaire, and four studies (20%) measured quality of life.

### 3.2. Quality Assessment of Articles

In this study, two researchers evaluated the quality of the literature using Cochrane’s RoB tool (Figure 2). Two studies (10%) showed a high risk of bias for blinding of participants and personnel; meanwhile, nine studies (45%) did not mention blinding of participants and personnel, meaning they could not be analyzed in this regard. One article (5%) showed a high risk of incomplete outcome data and selective reporting. When each study was observed individually (Figure 2B), no study showed a high risk of bias for more than one item; thus, all 20 studies were included in the analysis.

### 3.3. Effect Size Estimation

#### 3.3.1. Effects of Aquatic Exercise on Pain among Patients with Osteoarthritis

Of the 13 studies that measured pain, 10 included a control group that did not receive any treatment and presented pain scores in the form of means and standard deviations [14,22,26,27,29,31,33,34,37,39]; these 10 studies were selected for a meta-analysis (Figure 3A). Patients with osteoarthritis who performed aquatic exercise experienced a 0.61-point decrease in pain (*n* = 756; MD = −0.61; 95% CI: −0.90–−0.32) and the effect size differed significantly between the treatment group and the control group (Z = 4.09, *p* < 0.001). However, moderate heterogeneity was observed (I^2^ = 68%), and a subgroup analysis was performed to assess the cause of the heterogeneity. The 10 studies included in this analysis showed differences regarding participants. Eight studies were performed on subjects with osteoarthritis of the lower extremities, such as the hip and knee. The other two studies recruited patients with any form of osteoarthritis. Aquatic exercise has been found to be more effective in patients with osteoarthritis of the lower extremities [7]; therefore, we felt that it would be meaningful to compare the pain-related changes reported by the studies involving patients with osteoarthritis of the lower extremities with those reported by the studies featuring patients with osteoarthritis of the fingers or vertebrae. In the eight studies conducted on patients with osteoarthritis of the lower extremities, pain was reduced by 0.78 points (*n* = 388; MD = −0.78; 95% CI: −1.03–−0.52) when compared to the control group, the effect size differed significantly between the treatment group and the control group (Z = 9.82, *p* < 0.001), and low heterogeneity was observed (I^2^ = 29%). In the two studies that did not limit the focus to a particular body location, pain was reduced by 0.11 points (*n* = 368; MD = −0.11; 95% CI: −0.33–0.11) when compared to the control group, the effect size did not significantly differ between the treatment group and the control group, and no heterogeneity was observed (I^2^ = 0%).

Among the studies that measured pain and presented pain scores in the form of means and standard deviations, a meta-analysis was performed on six studies that included a control group that performed land-based exercise [23,26,27,31,34,38] (Figure 3B). This showed that pain in patients who performed aquatic exercise was reduced by 0.28 points (*n* = 315; MD = −0.28; 95% CI: −0.50–−0.05) when compared to the group that performed land-based exercise. The effect size differed significantly between the aquatic exercise group and the land-based exercise group (Z = 2.43, *p* = 0.001), and heterogeneity was not observed (I^2^ = 0%).

#### 3.3.2. Effects of Aquatic Exercise on the Quality of Life of Patients with Osteoarthritis

Among the four studies that measured quality of life, three included a control group that did not receive any treatment and presented the quality-of-life score in the form of means and standard deviations; these three studies [14,22,34] were selected for a meta-analysis (Figure 3A). This showed that the quality of life of the patients who performed aquatic exercise improved by 0.77 points (*n* = 279; MD = −0.77; 95% CI: −1.38–−0.15) when compared to the control group. The effect size differed significantly between the aquatic exercise group and the land-based exercise group (Z = 2.44, *p* = 0.001); however, high heterogeneity was observed (I^2^ = 80%).

#### 3.3.3. Effects of Aquatic Exercise on Joint Dysfunction among Patients with Osteoarthritis

Among the 20 studies that measured joint dysfunction in patients with osteoarthritis, seven included a control group that did not receive any treatment, assessed joint dysfunction through self-reported questionnaires, and presented the joint dysfunction scores in the form of means and standard deviations; these seven studies [14,26,27,33,34,37,39] were selected for a meta-analysis (Figure 3A). This showed that the joint dysfunction score was 0.34 points lower in the aquatic exercise group (*n* = 279; MD = −0.77; 95% CI: −1.38–−0.15) when compared to the control group. The effect size differed significantly between the two groups (Z = 3.58, *p* < 0.001), and heterogeneity was not observed (I^2^ = 0%).

Meanwhile, when considering studies that featured a land-based exercise group, the aquatic exercise group [26,27,34] showed a 0.14-point lower dysfunction score (*n* = 204; MD = −0.14; 95% CI: −0.42–0.13). The effect size did not significantly differ between the two groups, and heterogeneity was not observed (I^2^ = 0%; Figure 3B).

## 4. Discussion

This study concerned a systematic literature review of RCTs with the aim of assessing the effects of aquatic exercise on patients with osteoarthritis in terms of pain, joint function and quality of life. Twenty studies, selected based on specific inclusion and exclusion criteria, were qualitatively analyzed, and qualitative meta-analyses were performed on 12 of these studies that presented outcome variables, including pain, quality of life, and joint dysfunction, in the form of means and standard deviations.

The methods, session length, and program duration of the interventions varied across the 20 studies included in the analysis. Thus, it was difficult to analyze their effects. However, the aquatic exercises were generally performed in water that was 32–34 °C. Further, most studies concurrently included weight training and cardiovascular exercise, such as walking, running and weightlifting. Most studies administered 12-week programs featuring sessions lasting 50–60 min. The interventions showed beneficial effects on pain, quality of life and joint function. It is known that aquatic exercise, balneotherapy (which involves the use of mineral water), spa therapy (focusing on water temperature and flow) and Kneipp hydrotherapy (using water temperature and pressure) show positive effects for patients with osteoarthritis [19]. The results of the current study are consistent with these existing findings.

A meta-analysis was performed to assess the effects of aquatic exercise on pain reduction in patients with osteoarthritis. In most of the analyzed studies, pain was subjectively assessed using self-report questionnaires and visual analog scales. The meta-analysis showed that pain was significantly reduced in the aquatic exercise groups when compared to control groups that did not receive any treatment. Notably, the effect size differed depending on the site of the patients’ osteoarthritis. Pain significantly decreased in patients with hip and/or knee osteoarthritis; however, pain was not significantly reduced in studies that recruited patients with any form of osteoarthritis (Figure 3A). These results suggest that aquatic exercise can alleviate pain, especially among patients with osteoarthritis in their lower extremities. Strengthening the muscles around the joints through exercise is likely to increase joint function, contributing to pain alleviation [8,10]. However, only two studies included in the current study’s meta-analysis were conducted on patients with upper extremity osteoarthritis. Therefore, further studies assessing the causal relationship in this regard are necessary.

Moreover, aquatic exercise was also determined to be more effective than land-based exercise for reducing pain (Figure 3B). Previous studies have suggested that water temperature and other characteristics such as waves and pressure relax the muscles and soothe nerve endings, thereby reducing pain [40]. In addition, perceived pain reduction and muscle relaxation facilitate joint movement and may help to strengthen the muscles [11,15]. Therefore, this finding underlines the above finding that aquatic exercise may represent an effective intervention method for reducing pain and symptoms in patients with osteoarthritis.

A meta-analysis was performed on three studies that assessed the effects of aquatic exercise on the quality of life of patients with osteoarthritis. Quality of life is a multi-domain concept comprising physical, mental, social and spiritual aspects, and indicates one’s general well-being [41]. Osteoarthritis, which is accompanied by functional limitation of the joints, leads to pain in the affected joints, physical disability and decreased quality of life [42,43]. However, only three of the 20 studies included in the analysis assessed quality of life; therefore, future studies of the quality of life of patients with osteoarthritis are necessary. The durations of the aquatic exercise interventions were found to influence their effect on quality of life. When compared to control groups that did not receive any treatment, quality of life score increased as the intervention duration increased (Figure 3A). Studies included in the analysis administered six-week, eight-week and 20-week programs, and it may be difficult to make definitive conclusions based on the small number of studies in question. However, these findings suggest that continuous, rather than temporary, aquatic exercise is necessary for patients with osteoarthritis. Note, however, that aquatic exercise can only be performed in pools of appropriate depth and water temperature; therefore, the accessibility of such facilities and the cost of their use should be considered, as these variables may create difficulties regarding performing such exercises regularly [3]. Based on the above findings, a social consensus must be formed, and public medical support must be expanded. In addition, awareness of the need for aquatic exercise decreases over time, leading to reduced performance rate [44]. Thus, behavioral strategies and education that reinforce factors that can increase motivation and, thus, the efficacy of the interventions, must be identified and administered [44].

Aquatic exercise showed beneficial effects for the joint function of patients with osteoarthritis. A meta-analysis revealed significant differences between the aquatic exercise group and the control group in this regard; however, no such significant differences were observed between the aquatic exercise group and the land-based exercise group. Exercise, regardless of setting, may increase joint strength and range of movement, leading to improved joint function [3,8]. However, the incidence rate of osteoarthritis is high in overweight patients, and the buoyancy of water supports body weight, thereby decreasing the impacts on joints; this helps to alleviate pain and facilitate movement [3,11]. This represents a possible reason aquatic exercise is more suitable than land-based exercise for improving joint function. However, as mentioned above, aquatic exercise has limitations in regard to space, time and cost. Thus, appropriate measures to overcome these limitations must be considered. A possible solution would be locally based intervention methods that use the unique characteristics of water to produce similar effects to aquatic exercise.

This study performed a systemic literature review and meta-analyses of RCTs in which aquatic exercise-based interventions were administered to patients with osteoarthritis. This study is significant as, when compared to previous studies that individually verified the effects of such interventions, the present study verified the effects of these interventions in an integrated and scientific manner. However, despite the RCT design of the studies, when the quality of the literature was evaluated risks of bias in relation to random sequence generation, allocation concealment, blinding of participants and personnel and blinding of outcome assessment were observed. Furthermore, a limited number of studies assessing quality of life was included in the meta-analysis. Therefore, future studies that feature more comprehensive approaches and designs are required. In addition, the explanations of the aquatic exercise programs lacked detail. No explanations were given regarding the evidence base for choosing the program durations, numbers of sessions per week, and the durations of each session. Therefore, in terms nursing research, to provide strong evidence for the use of aquatic exercise as a nursing intervention, it is critical to increase the level of existing evidence by applying more rigorous and sophisticated research methods. In terms of nursing practice, it is thought that it will contribute to improving the quality of life and maintaining health by using it as a supplementary alternative therapy to alleviate pain and function in osteoarthritis patients. Finally, PROSPERO is an international database of prospectively registered systematic reviews with a health-related outcome. However, the limitation of this study is that it is not registered with PROSPERO.

In conclusion, underwater exercise is effective in reducing pain and improving function and quality of life in patients with osteoarthritis. Therefore, aquatic therapy can be used not only with medication, but also with other non-pharma and non-surgical interventions such as land-based exercise [45], manual therapy [46], knee bracing [47] and physical modalities [48].

## 5. Conclusions

This study systemically investigated, through assessment of RCTs, the effects of aquatic exercise on pain, quality of life and dysfunction in patients with osteoarthritis. As a result, aquatic exercise was found to alleviate pain, increase quality of life and reduce dysfunction in such patients. However, as the optimal program duration, session frequency and session duration for aquatic exercise have not yet been determined [3], future studies that analyze these effects are necessary. Furthermore, measures to overcome the space-, time- and cost-related limitations associated with aquatic exercise must be sought.

## Figures and Tables

**Figure 1 healthcare-10-00560-f001:**
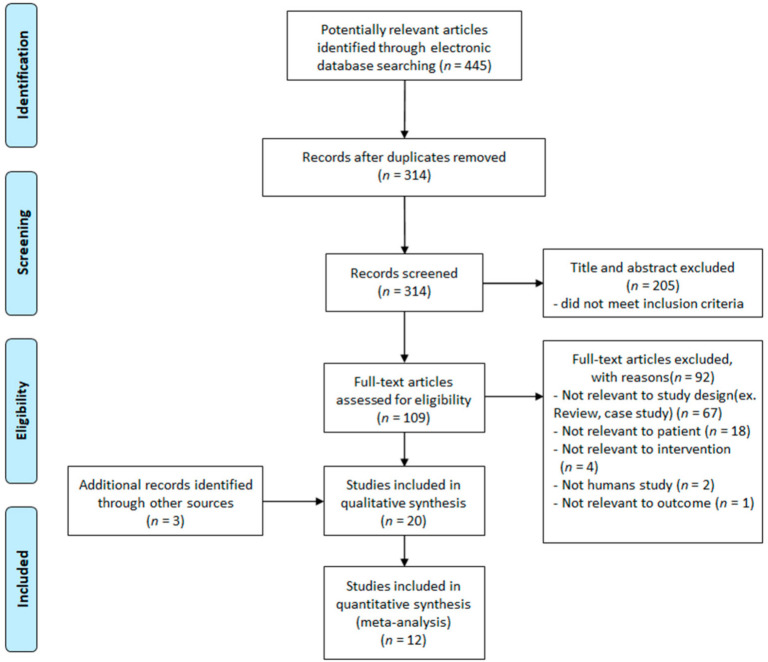
Study flow chart of study selection process.

**Figure 2 healthcare-10-00560-f002:**
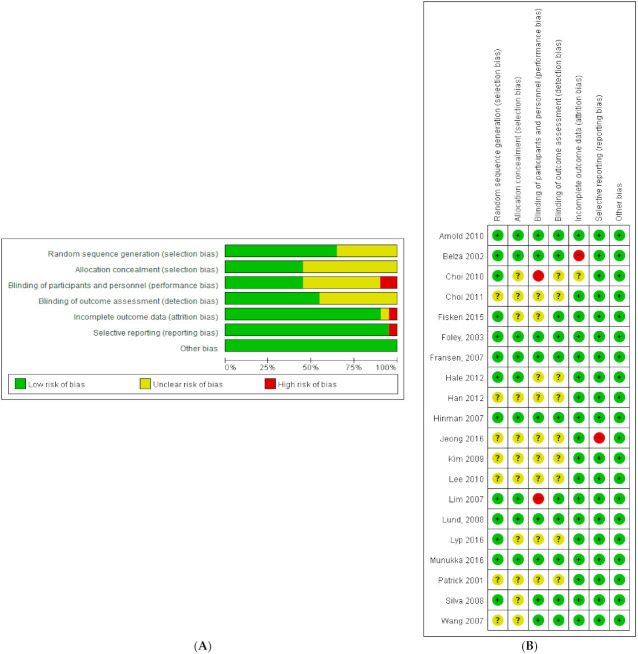
Reviewing authors’ judgments about each risk of bias item presented as percentages across all included studies. (**A**) Risk of bias graph and (**B**) risk of bias summary [14,21,22,23,24,25,26,27,28,29,30,31,32,33,34,35,36,37,38,39].

**Figure 3 healthcare-10-00560-f003:**
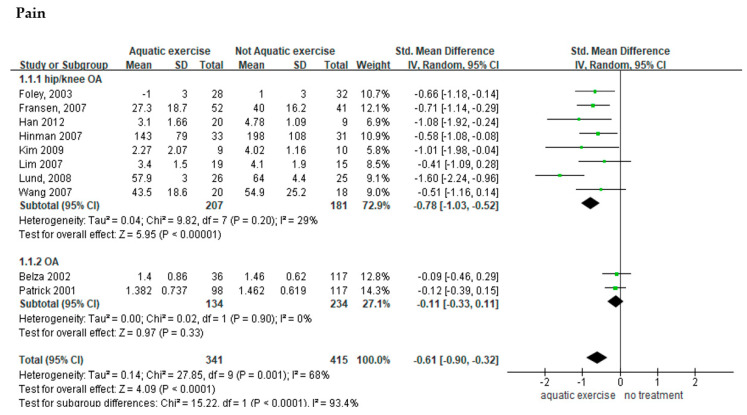
Forest plot of the effects of aquatic exercise for pain, quality of life and function in osteoarthritis patients. Effects of aquatic exercise versus (**A**) no treatment and (**B**) land exercise. M = mean; SD = standard deviation; CI = confidence interval; df = degrees of freedom; MD = mean difference; SMD = standardized mean difference [14,22,23,26,27,29,31,33,34,37,38,39].

**Table 1 healthcare-10-00560-t001:** Summary of randomized controlled trials examining aquatic exercise in osteoarthritis patients.

First Author (Years)	Sample Size Age	Intervention Group (Regime)	Control Group (Regime)	Aquatic Exercise	Main Outcome Measures	Intergroup Difference	Author’s Conclusion
Arnold (2010)	79 adults with hip OA, over 65 years	(1) Aquatic exercise, Chest depth (*n* = 26) (2) Aquatic exercise and Education, Chest depth (*n* = 28) * walking, stretching, jump, dumbbell	(3) No treatment (*n* = 25)	(1) Duration 45 min, 2 days/week, 11-week (2) Duration 30-min education + 45-min aquatic exercise 2 days/week, 11-week	(1) BBSm (2) 6-min Walk test (3) 30-s chair stand (4) ABC-Q (5) TUG_cog_	(1) NS (2) NS (3) *p* = 0.022 (4) *p* = 0.047 (5) NS	“The combination of aquatic exercise and education was effective in improving fall risk factors in older adults with arthritis.”
Belza (2002)	222 adults with OA, 55~75 years	(1) Aquatic exercise, adherence, 85~92 °F (*n* = 36) * upper- and lower-body activities	(2) Aquatic exercise, non-adherence, 85~92 °F (*n* = 65) (3) No treatment (*n* = 121)	(1) Duration 1 h, ≥2 days/week, 20-week (2) Duration 1 h < 2 days/week, 20-week	(1) QWB (2) HAQ (3) Pain-VAS (4) CES-D (5) QOL	(1) *p* = 0.02 (2) *p* = 0.02 (3) NS (4) *p* =0.035 (5) *p* < 0.01	“When analyzed for level of participation, exercise benefits adults with osteoarthritis. Improved methods are needed to enhance adherence, with increased attention…”
Chi (2010)	29 women with OA, Avg. age 61 years	(1) Aquatic exercise, Chest depth (*n* = 13) * walking, stretching, running	(2) Stretching exercise (*n* = 16)	(1), (2) Duration 50 min, 2 days/week, 8-week	(1) Flexibility (2) Pain-VAS (3) QOL	(1) NS (2) NS (3) NS	“The 8-week stretching exercise program would significantly improve flexibility, pain control and QOL in patients with osteoarthritis.”
Choi (2011)	30 women with knee OA, over 65 years	(1) Aquatic exercise, Chest depth, 28~30 °C (*n* = 15) * walking, stretching, jump	(2) No treatment (*n* = 15)	(1) Duration 50 min, 5 days/week, 24-week	(1) Muscle function (2) T-score(pelvic) (3) GDS-K	(1) *p* < 0.05 (2) *p* < 0.001 (3) *p* < 0.001	“Aquatic exercise very effective on improving leg muscle function and T-score as well as depression.”
Fisken (2015)	25 adults with OA, over 60 years	(1) Aquatic exercise, Chest depth, 87 °F (*n* = 13) * aerobic, strength	(2) Seated-aquatic DVD exercise, 97.7 °F (*n* = 12)	(1) Duration 60 min, 2 days/week, 12-week (2) Duration 40 min, once/week, 12-week	(1) TUG (2) Step test (3) Sit-to-stand test (4) Handgrip (5) 400-m walk test (6) AIMS2-SF (7) FES-I	(1) NS (2) NS (3) NS (4) NS (5) NS (6) NS (7) *p* < 0.05	“Aqua fitness may offer a number of positive functional and psychosocial benefits for older adults with OA...”
Foley (2003)	35 adults with hip and knee OA, over 50 years	(1) Aquatic exercise, no information (*n* = 28) (2) Land-based exercise (*n* = 26) * walking, strength	(3) No treatment (*n* = 32)	(1), (2) Duration 30 min, 3 days/week, 6-week	(1) Walk speed (2) Muscle strength (3) WOMAC-pain (4) WOMAC-stiff. (5) WOMAC-func. (6) SF-12 PCS (7) SF-12 MCS	(1) B/C *p* = 0.009 (2) A/B, B/C *p* < 0.05 (3) NS (4) NS (5) NS (6) A/C *p* < 0.05 (7) NS	“Functional gains were achieved with both exercise programs compared with the control group.”
Fransen (2007)	145 adults with hip and knee OA, 59~85 years	(1) Aquatic exercise, Waist depth, 34 °C (*n* = 52) (2) Tai Chi (*n* = 52) * walking, stretching, strength, raise	(3) No treatment (*n* = 41)	(1), (2) Duration 60 min, 2 days/week, 12-week	(1) WOMAC-pain (2) WOMAC-func. (3) SF-12 PCS (4) SF-12 MCS (5) Depression (6) Anxiety (7) Stress (8) Up-and-Go (9) 50-foot walk time (10) Stair climb	(1) A/C *p* < 0.05 (2) A/C, B/C *p* < 0.05 (3) A/C, B/C *p* < 0.05 (4) NS (5) NS (6) NS (7) NS (8) A/C, B/C *p* < 0.05 (9) A/C *p* < 0.05 (10) A/C, B/C *p* < 0.05	“Access to either hydrotherapy or Tai Chi classes can provide large and sustained improvements in physical function for many older, sedentary individuals with chronic hip or knee OA.”
Hale (2012)	35 adults with hip and knee OA, over 65 years	(1) Aquatic exercise, Chest depth, 28 °C (*n* = 20) * walking, stretching, jump, dumbbell	(2) No treatment, time-matched computer training program (*n* = 15)	(1), (2) Duration 60 min, 2 days/week, 12-week	(1) PPA (2) Step test (3) TUG (4) WOMAC (5) AIMS2-SF (6) ABC-Q	(1) NS (2) NS (3) NS (4) NS (5) NS (6) NS	“Water-based exercise did not reduce falls risk in our sample compared with attending a computer skills training class.”
Han (2012)	29 women with hip and knee OA, 65~70 years	(1) Aquatic exercise (*n* = 10) (2) Swimming exercise (*n* = 10) * walking, stretching, jump	(3) No treatment (*n* = 9)	(1), (2) Duration 50 min, 3 days/week, 12-week	(1) In body (2) Step length (3) Gait cadence (4) Speed (5) Pain-VAS	(1) NS (2) NS (3) NS (4) *p* < 0.01 (5) *p* < 0.05	“Aquarobic group showed significant increase of lean body mass, and significant decrease body fat, BMI, pain…”
Hinman (2007)	64 adults with hip and knee OA, over 60 years	(1) Aquatic exercise, Chest depth, 34 °C (*n* = 33) * walking, stretching, hitching, raise	(2) No treatment (*n* = 31)	(1) Duration 45~60 min, 2 days/week, 6-week	(1) Pain-VAS (2) WOMAC-pain (3) WOMAC-stiff. (4) WOMAC-func. (5) QOL (6) PASE (7) Muscle strength (8) Step test (9) TUG (10) 6-min Walk test	(1) *p* < 0.05 (2) *p* < 0.001 (3) *p* < 0.01 (4) *p* < 0.001 (5) *p* < 0.05 (6) NS (7) *p* < 0.05 (8) NS (9) NS (10) *p* < 0.01	“Compared with no intervention, a 6-week program of aquatic physical therapy resulted in significantly less pain and improved physical function, strength, and quality of life.”
Jeong (2016)	30 women with knee OA, over 65 years	(1) Aquatic exercise, Chest depth, 29~30 °C (*n* = 10) (2) Land-based exercise, Nordic walking (*n* = 10) * walking, stretching, raise	(3) No treatment (*n* = 10)	(1), (2) Duration 60 min, 4 days/week, 8-week	(1) WOMAC (2) Arm curl test (3) 30-s chair stand (4) Back scratch (5) Chair sit-reach (6) 2.44-m-TUG (7) 6-min Walk test	(1) *p* < 0.001 (2) *p* < 0.001 (3) NS (4) *p* < 0.001 (5) *p* < 0.001 (6) *p* < 0.001 (7) NS	“Walking exercise in the water and walking exercise on the ground are positive exercises that can promote health for the elderly women with degenerative arthritis….”
Kim (2009)	28 women with knee OA, over 65 years	(1) Aquatic exercise, Chest depth, 28 °C (*n* = 9) (2) Land-based exercise, Nordic walking (*n* = 9) * walking, stretching, jump, raise	(3) No treatment (*n* = 10)	(1), (2) Duration 60 min, 4 days/week, 8-week	(1) Pain-VAS (2) Free Oxygen Radical (3) TAC	(1) *p* < 0.01 (2) *p* < 0.01 (3) *p* < 0.01	“…aquatic exercise in combination including decreases of ROS, are safe and moderately effective…”
Lee (2010)	29 women with knee OA, 60~70 years	(1) Aquatic exercise (*n* = 10) (2) Swimming exercise (*n* = 10) * walking, stretching, running, jump, raise	(3) No treatment (*n* = 9)	(1), (2) Duration 50 min, 3 days/week, 12-week	(1) Power (2) Isometric (3) Leg length (4) Knee joint distance (5) ROM	(1) NS (2) NS (3) NS (4) NS (5) NS	“Aquarobic exercise more effective than swimming on leg muscular strength and ROM for degenerative arthritis.”
Lim (2007)	34 adults with knee OA, Avg. 65.9(A)/62.1(B) years	(1) Aquatic exercise, Chest depth, 32 °C (*n* = 19) * no information	(2) No treatment (*n* = 15)	(A) Duration 30 min, 3 days/week, 8-week	(1) Pain-BPI (2) Interference-BPI (3) WOMAC (4) SF-36 PCS (5) SF-36 MCS (6) body mass (7) Body fat mass (8) Waist-hip ratio	(1) *p* < 0.05 (2) *p* < 0.05 (3) *p* = 0.001 (4) NS (5) NS (6) NS (7) NS (8) NS	“Aquatic exercise is an effective tool for obese patients who have a difficulty in active exercise due to combined knee osteoarthritis.”
Lund (2008)	192 adults with knee OA, Avg. 65.0(A)/70.0(B)/ 68.0(C) years	(1) Aquatic exercise, Chest depth, 33.5 °C (*n* = 26) (2) Land-based exercise (*n* = 20) * walking, stretching, running	(3) No treatment (*n* = 25)	(1) Duration 50 min, 2 days/week, 8-week	(1) Pain-VAS (2) KOOS-symp. (3) KOOS-pain (4) KOOS-ADL (5) KOOS-sport (6) KOOS-QOL (7) Muscle strength (8) Balance	(1) NS (2) NS (3) NS (4) NS (5) NS (6) NS (7) NS (8) NS	“Only land-based exercise showed some improvement in pain and muscle strength compared with the control group…”
Lyp (2016)	192 adults with hip OA, 48~82 years	(1) After THR, Aquatic exercise + Kinesitherapy + low-frequency magnetic field, Chest depth, 34 °C (*n* = 32) (2) Aquatic exercise + Kinesitherapy + low-frequency magnetic field, Chest depth, 34 °C (*n* = 32) * walking, stretching	(3) After THR, Kinesitherapy + low-frequency magnetic field (*n* = 32) (D) Kinesitherapy + low-frequency magnetic field (*n* = 32) (E) After THR, No treatment (*n* = 32) (F) No treatment (*n* = 32)	(1), (3) Duration 30 min, 5 days/week, 4-week	(1) Pain-VAS (2) Pain-Laitinen scale (3) HAROM (4) Strength of hip joint	(1) *p* < 0.001 (2) *p* < 0.01 (3) −1. A/B/E NS (3) −2. C/D/F NS (4) −1. A/B/E NS (4) −2. C/D/F NS	“Inclusion of water exercises in a rehabilitation program can reduce the use of medicines in patient with OA and after THR.”
Munukka (2016)	84 women with knee OA, 60~68 years	(1) Aquatic exercise, Chest depth, 30~32 °C (*n* = 42) * resistance ROM	(2) No treatment (*n* = 42)	(1) Duration 60 min, 3 days/week, 16-week	(1) T2 relaxation time (2) dGEMRIC (3) VO_2_ peak (4) KOOS	(1) *p* < 0.05 (2) *p* < 0.05 (3) *p* = 0.01 (4) NS	“Additionally, aquatic resistance training of sufficient intensity low risk of harm amongst women with mild knee OA.”
Patrick (2001)	225 adults with OA, 55~75 years	(1) Aquatic exercise: adherence, Chest depth, 85~92 °F (*n* = 36) (2) Aquatic exercise: non-adherence, Chest depth, 85~92 °F (*n* = 68) * upper- and lower-body activities	(3) No treatment (*n* = 121)	(1) Duration 45~60 min, 2 days/week, 20-week (2) Duration 45~60 min, <2 days/week, 20-week	(1) QWB (2) Health desirability (3) HAQ-disability (4) HAQ-pain (5) PQOL-physical (6) CES-D	(1) NS (2) *p* < 0.05 (3) *p* < 0.05 (4) NS (5) *p* < 0.01 (6) NS	“Aquatic exercisers reported equal (QWB) or better (CHDR, HAQ, PQOL) health-related quality of life compared with controls.”
Silva (2008)	64 adults with knee OA, Avg. age 59 years	(1) Aquatic exercise, Chest depth, 32 °C (*n* = 32) * stretching, strengthening	(2) Land-based exercise (*n* = 32)	(1), (2) Duration 50 min, 3 days/week, 18-week	(1) Pain-VAS (2) Lequesne index (3) WOMAC (4) After 50FWT Pain-VAS (5) Walking time-fast (6) Walking time	(1) NS (2) NS (3) NS (4) *p* < 0.01 (5) NS (6) NS	“Water-based exercises are a suitable and effective alternative for the management of OA of the knee.”
Wang (2007)	38 adults with hip and knee OA, Avg. age 69.3(I)/62.7(C) years	(1) Aquatic exercise, Chest depth, 30~32 °C (*n* = 20) * walking, stretching, lift	(2) No treatment (*n* = 18)	(1) Duration 50 min, 3 days/week, 12-week	(1) Flexibility (2) Strength-knee (3) Strength-hip (4) Physical function (5) Pain-VAS	(1) *p* < 0.05 (2) *p* < 0.05 (3) *p* < 0.05 (4) NS (5) NS	“Beneficial short-term effects of aquatic exercise were found in adults with osteoarthritis of the hip or knee.”

BBSm = Berg Balance Scale-modified; ABC-Q = Activities and Balance Confidence Questionnaire; TUG_cog_ = Time Up-and-Go test, Dual-Task cognitive; QWE = Quality of Well-Being; HAQ = Health Assessment Questionnaire; VAS = Visual Analog Scale; CES-D = Center for Epidemiological Studies-Depression scale; QOL = Quality of Life; GDS-K = Geriatric Depression Scale in Korea. TUG = Time Up-and-Go test; AIMS2-SF = Arthritis Impact Measurement Scale-Short Form; FES-I = Falls Efficacy Scale-International; WOMAC = Western Ontario and McMaster Universities OA Index; PCS = Physical Component Scale; MCS = Mental Component Scale. PPA = Physiological Profile Assessment; TUG = Time Up-and-Go test; PASE = Physical Activity Scale for the Elderly; TAC = Total antioxidant capacity; ROM = Range of Motion. BPI = Brief Pain Inventory; KOOS = Knee injury and Osteoarthritis Outcome Score (self-assessed impact of OA pain, other symptoms, activities of daily living, sport, quality of life); THR = Total Hip Replacement; HAROM = Hip Active Ranges of Motion; dGEMRIC = delayed Gadolinium-Enhanced Magnetic Resonance Imaging of Cartilage; QWB = Quality of Well-Being scale; PQOL = Perceived Quality of Life scale; CES-D = Center for Epidemiological Studies-Depression scale; FWT = Feet Walk Test. * = Exercise type. NS = not significant.

## Data Availability

Data sharing is not applicable to this article.

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
