# Peer review of "Effects of Aquatic Exercises for Patients with Osteoarthritis: Systematic Review with Meta-Analysis"

_healthcare, 2022, doi:10.3390/healthcare10030560_

Round 1

Reviewer 1 Report

BRIEF SUMMARY

This was a systematic review with a meta-analysis, to determine the effects of aquatic-exercise-based interventions on pain, quality of life, and joint dysfunction among patients with osteoarthritis. (OA). Authors included 20 studies and reported that aquatic therapy is effective in improving pain, quality of life and joint dysfunction in people with OA compared to a control group.

This is a very detailed and technically sound paper with informative figures and tables. Overall, I found the topic timely and clinically important. My main concern is quality of reporting in the methods which makes it difficult to fully understand what and how things were done. Before it can be published I suggest authors to consider my points below.

SPECIFIC COMMENTS

TITLE

Could you please rephrase the title to better represent what was done? For example: Effects of Aquatic Exercises for Patients with Osteoarthritis: A systematic review with meta-analysis.

ABSTRACT

Line 8: it seems you only included papers published in Korean whereas in the methods you specify that also English papers were included (Line 106). Please correct this inconsitency throughout the paper.

Line 10-12: please rephrase, it is dificult to understand this sentence.

INTRODUCTION

Line 27-30 – low grade systemic inflammation is also an important sign in OA and should be acknowledged:

https://pubmed.ncbi.nlm.nih.gov/28929165/

https://pubmed.ncbi.nlm.nih.gov/9263139/

In the current form, it is quite difficult to figure out from the information flow in the introduction, why it is important to study this, who will benefit from it, and what is the added value of this paper to current knowledge. Please clarify.

Line 67-71: please delete; it seems redundant as you introduce this in the methods anyway.

METHODS

http://www.prisma-statement.org/ . Please report according to PRISMA checklist and include it with your submission as a supplementary file.

Please explain why the study protocol has not been published in PROSPERO prior to study commencement.

Re outcomes: Please provide information which outcome data did you use (baseline, follow-up or change data) for meta-analysis? Alos, if a study measured and reported pain using two outcome measures for example, NRS and WOMAC, which did you use in your meta-analysis?

Please include as a table a search strategy for one example search engine in the main text Also, supplementary material should include an actual search strategy for each engine.

RESULTS

Very well structured.

DISCUSSION

The discussion/introduction lacks reference to other reviews in the field. If this is the first review of this kind in the field, you should acknowledge this in the intro. If other reviews are present, you should also introduce them and discuss your results with previous reviews.

Please provide infomation how your results will impact research and/or clinical practise.

Please disucsss how the generalizabity of the results to the wider OA population.

CONCLUSIONS

Line 422-423: This statement should not belong to conclusions as you have not done anything with drug therapy. Instead, I believe that discussion should have included a seperate paragraph discussing that aquatic therapy could be used in conjuction with drug therapy but also with other non-pharma and non-surgical interventions such as land-based exercise (https://www.ncbi.nlm.nih.gov/pmc/articles/PMC3635671/), manual therapy (https://www.ncbi.nlm.nih.gov/pmc/articles/PMC2597887/ ), knee bracing (https://pubmed.ncbi.nlm.nih.gov/30099859/ ), physical modalities ( https://pubmed.ncbi.nlm.nih.gov/25162407/ ) etc.

Author Response

Thank you for your opinion. 

Please refer to the following file for corrections.

Reviewer 2 Report

As the growing number of OA affected patients, the study analyses important medical problem. Exercises therapy is a well-documented treatment modality for improving patients QoL . The authors analyze current literature aiming to recognize the role of aquatic exercises in OA management. The analysis is well organized and written. Additionally few years have passed since similar topic was analysed (DOI: 10.1002/14651858.CD005523.pub3  and doi: 10.1097/MD.0000000000013823.)

The only issue is that some newer trials are missing in the results of search string. Maybe, authors should consider to add recent trials to the analysis:

doi: 10.1016/j.apmr.2019.12.023. Epub 2020

DOI: 10.1111/sms.13630

https://doi.org/10.1177/0269215517754240

DOI: 10.1016/j.joca.2017.02.800

Author Response

Thank you for your opinion.

Since the document search date for this study was until May 7, 2019, recent studies were not included. Including recent studies, there is a difficulty in requiring extensive revision of research results and discussions. We will conduct a systematic literature review including the latest research in the future.

Reviewer 3 Report

Thanks for the opportunity to review the manuscript entitled "Systematic Review of the Effects of Aquatic Exercises for 2 Patients with Osteoarthritis ".  The authors have comprehensively reviewed related trails in the related era. And the methodology and discussion was appropriate. Only a minor commend need to be addressed.

The systematic review  should further include trails between 2019 and 2022.

Author Response

(The authors gave the same response as above.)

Round 2

Reviewer 1 Report

I thank the authors for their revisions. Several comments are still not addressed:

Line 30-31: systemic inflammation is also an important sign in osteoarthritis. Please mention it here and cite the following papers:

https://pubmed.ncbi.nlm.nih.gov/28929165/

https://pubmed.ncbi.nlm.nih.gov/9263139/

Re PROSPERO: please acknowledge this in your limitations that the study has not been registered.

A short paragraph is needed in the discussion saying that aquatic therapy could be used in conjunction with drug therapy but also with other non-pharma and non-surgical interventions such as land-based exercise (https://www.ncbi.nlm.nih.gov/pmc/articles/PMC3635671/), manual therapy (https://www.ncbi.nlm.nih.gov/pmc/articles/PMC2597887/ ), knee bracing (https://pubmed.ncbi.nlm.nih.gov/30099859/ ), physical modalities ( https://pubmed.ncbi.nlm.nih.gov/25162407/ ). Please discuss it shorty and cite the papers I suggested.

Author Response

Thank you for your valuable review.

Please refer to the file for answers to the three comments.

This manuscript is a resubmission of an earlier submission. The following is a list of the peer review reports and author responses from that submission.